# Seasonal Dynamics of Soil Microbial Biomass C, N and P along an Altitudinal Gradient in Central Himalaya, India

Vijyeta Manral [1], Kiran Bargali [1], Surendra Singh Bargali [1,*], Himani Karki [1] and Ravi Kant Chaturvedi [2,*]

[1] Department of Botany, DSB Campus, Kumaun University, Nainital 263001, Uttarakhand, India
[2] Center for Integrative Conservation, Xishuangbanna Tropical Botanical Garden, Chinese Academy of Sciences, Menglun 666303, China
* Correspondence: surendrakiran@rediffmail.com (S.S.B.); ravi@xtbg.ac.cn (R.K.C.)

**Abstract:** This study was conducted in a temperate mixed oak–pine forest of Central Himalaya, India to (i) evaluate altitudinal and seasonal variations in the microbial biomass carbon (C), nitrogen (N) and phosphorus (P) and (ii) analyse the relationships between soil microbial biomass C, N and P and physico-chemical properties of soil. Three permanent plots were established in natural forest stands along an altitudinal gradient, three replicates were collected seasonally from each site, and microbial biomass (C, N and P) were determined by a fumigation extraction method. Microbial biomass C, N and P decreased significantly ($p < 0.01$, correlation coefficient 0.985, 0.963, 0.948, respectively) with increasing altitude having maximum values during rainy season and minimum values during winter season. Microbial biomass C, N and P showed positive correlations with silt particles, water holding capacity, bulk density, soil moisture, organic C, total N and P and negative correlations with sand particles, porosity and soil pH. Microbial biomass C was strongly associated with soil microbial N (r = 0.80, $p < 0.01$) and P (r = 0.89, $p < 0.01$) content and soil microbial biomass N and P also showed a strong linear relationship (r = 0.92, $p < 0.01$). Soil microbial biomass exhibited weak seasonality and was highly influenced by altitude and abiotic variables. The significantly high microbial C, N and P during the rainy season ($p < 0.01$) and low microbial biomass during the winter season may be due to higher immobilization of nutrients from decomposing litter by microbes as the decomposition rate of litter and microbial activity are at their peak during the rainy period. The microbial C:N ratio indicated that soil fertility is influenced by species composition. Our findings suggested that high microbial biomass and low C:N ratios during the rainy season could be considered a nutrient conservation strategy of temperate mixed oak–pine forest ecosystems.

**Keywords:** abiotic variables; altitude; immobilization; mineralization; mixed oak–pine forest

## 1. Introduction

Soil microbes play a critical role in carbon (C) and nutrient transformation in forest soils [1]. The transformation ability of microbial biomass helps in conversion of complex organic matter present in soil into inorganic compounds that can be reused by plants. As such, biomass is both a source and sink of nutrients (carbon, nitrogen, phosphorus and sulphur, etc.) contained in organic matter. In soils, the decomposition of organic matter by microorganisms provides nutrients required by land plants [2,3]. It is the prime location of the majority of biological activity in soil and comprises about 2–3% of total organic carbon, and can be considered as a labile pool of essential plant nutrients such as nitrogen (N), phosphorus (P) and sulphur (S), which are held in a form largely protected from loss due to leaching or fixation [2].

In the cycling of important nutrients such as C, N and P, microbial activity is associated with the mineralization of these nutrients [4]. The soil microbial biomass plays an important role in soil processes such as N mineralization, and acts as a sensitive bio-indicator of on-going climatic changes [5]. Moreover, the microbial biomass can easily react to changes in

nutrients, moisture, temperature and the type and amount of soil organic matter within a turnover time of less than a year [6]. Thus, soil microbial biomass acts as indicator of changes in nutrient status [7,8], vegetative composition [9], and climatic conditions [10].

Forest vegetation affects the microbial processes of carbon and nitrogen cycling due to differences in quality and quantity of litter, root exudates, and soil properties [11–15]. Tree species have an impact on soil fertility and microbial community composition, which in turn can affect the soil microbial biomass and microbial efficiency in carbon utilization [16–18]. The ratio of microbial biomass C to microbial biomass N is an indicator of the structure and the state of a microbial community. Subsequently, a high microbial biomass C to microbial biomass N ratio indicates that the microbial biomass contains a high proportion of fungi, whereas a low value suggests that bacteria predominate in the microbial populations [19].

The activity of microorganisms in organic matter decomposition determines the mineralization and immobilization of nutrients, which affects the availability of nutrients in soil for plant growth [20]. Moreover, patterns of availability of limiting nutrients such as nitrogen and phosphorus in space and time influence plant communities and ecosystem functioning [21–24].

We conducted a field study in a natural mixed oak–pine temperate forest in Central Himalaya, India, with the following objectives: (i) to evaluate the altitudinal and seasonal variations in microbial biomass C, N and P; and (ii) to analyse the relationship between soil microbial biomass C, N and P and physico-chemical properties of soil. Two hypothesis were tested: (i) that altitude plays a major role in determining microbial biomass C, N and P by modifying climatic and soil characteristics, and (ii) that at the same altitude, microbial biomass C, N and P are affected by seasonal variations due to changes in microbial activity.

The purpose of the study was to provide information on altitudinal and seasonal variation in soil microbial biomass C, N and P in an annual cycle in a mixed oak–pine forest ecosystem to understand the role of microbes in annual nutrient flux.

## 2. Materials and Methods

### 2.1. Site Description

This study was conducted in the Kumaun Himalayan region near Nainital town in Uttarakhand State, India (29°19′29°28′ N latitude and 79°22′79°38′ E longitude). Three natural forest sites dominated by Chir pine (*Pinus roxburghii* Sarg.)  and Banj oak (*Quercus leucotrichophora* A. Camus) were selected, covering a 200 m vertical transition zone with elevations around 1500 m, hill base (HB), 1600 m, hill slope (HS) and 1700 m, hill top (HT), having similar topographic and environmental factors, such as slope, aspect and forest type. The dominant shrub species were *Hypericum cernuum* and *Indigofera heterantha* at the HB; *Rumex hastatus* and *Pyracantha crenulata* at the HS; and *Rubus ellipticus* and *Berberis asiatica* at the HT.

The mean monthly minimum temperature varied from 4 °C (January) to 17 °C (June) while the mean monthly maximum temperature ranged from 11 °C (February) to 26 °C (June) during the study period. The total precipitation was recorded at about 2527 mm and ranged from nil precipitation (November) to 842 mm (July) (Figure 1).

Geologically the study sites were located in the lesser Himalayan zone. According to Valdiya [25], the rocks are a complex mixture of sedimentary, low grade metamorphosed and igneous rocks and belong to the Krol series of the lesser Himalaya. A sequence of limestones, grey and greenish-grey and purple slates, siltstones, and in the upper part massive dolomites that follow the Blaini without a perceptible break was named by Medlicott [26] as the Krol series after the Krol mountains. The Baliani rock consists of conglomerates and siltstones. The Krol formation consists predominantly of carbonates, limestones, marl and slates in the lower part and dolomites on the upper part [25].

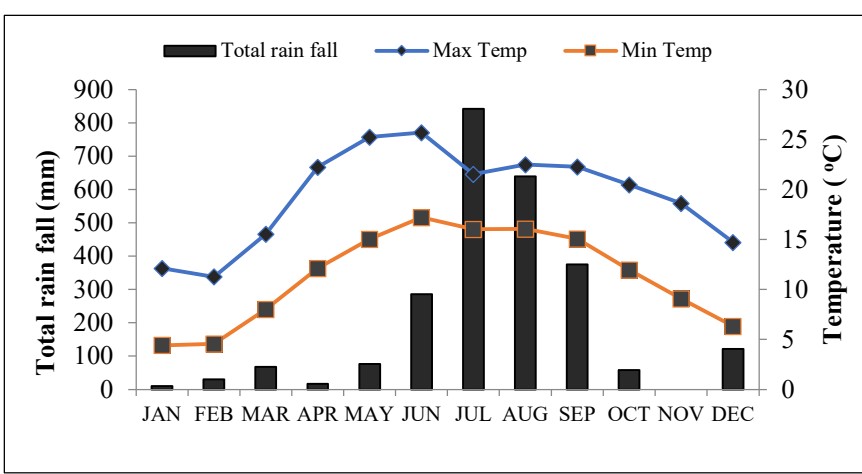

**Figure 1.** Meteorological data during the study period (Source: ARIES, Nainital).

## 2.2. Experimental Design

Soil was sampled from surface layer (0–15 cm) because most of the microbial activity is confined to this region [27,28]. Soil samples were collected randomly from each forest stand in rainy, winter and summer seasons in triplicate by digging soil monoliths (10 cm long × 10 cm wide × 15 cm deep). In order to randomize the samples, we kept a minimum distance between the soil samples at each site of at least 50 m. The collected soil samples were kept in a dry ice box and were brought to the laboratory for analysis of physical, chemical and biological properties. Soil texture was analysed following the Indian standard [29], and soil moisture using a gravimetric method. Soil pH was determined (1:5 water suspension) using a pH meter. The bulk density of the soil $(g\,cm^{-3})$ was calculated using mass and volume. Pore space was calculated using the bulk and particle density. Soil organic C was estimated by the rapid titration method [30,31], total N by the micro-Kjeldhal digestion technique [32] and total P using a spectrophotometer [31].

## 2.3. Analyses of Soil Microbial Biomass

The fresh soil samples were divided into two equal halves/sets; one set was immediately extracted (0.5 M $K_2SO_4$ for microbial C and N or 0.5 M $NaHCO_3$ for microbial P), and the other set was fumigated with chloroform and then extracted [33,34]. Soil microbial biomass C (SMBC) was determined by modified the Walkley–Black method and calculated following Jenkinson and Ladd [35]:

$$\text{Microbial C} = \text{KEC} \times 0.45$$

Microbial biomass N was determined by the micro-Kjeldahl method and calculated by Brookes et al. [34]:

$$\text{Microbial N} = \text{KEN} \times 0.54$$

and microbial biomass P was determined by the ammonium molybdate stannous chloride method and calculated by Brookes et al. [33]:

$$\text{Microbial P} = \text{KEP} \times 0.40$$

where, KEC, KEN and KEP are the differences between C, N and P extracted from fumigated and non-fumigated soils.

## 2.4. Data Processing

The raw data was statistically analysed to check the significance between the studied parameters using SPSS software (version 25). One-way analysis of variance (ANOVA) following Duncan's post hoc test was performed to study the variations in soil physico-

chemical properties across the sites. Two-way ANOVA was performed to analyse the effect of altitude, season and their interactions on soil microbial biomass C, N and P. Pearson's correlation analysis was conducted to determine relationship between measured parameters. Excel statistical software was used for the principal component analysis (PCA).

## 3. Results

### 3.1. Soil Characteristics

The soil was sandy loam with 70 to 78% sand, 10 to 13% clay and 12 to 18% silt. Soil moisture ranged from 11 to 22%, soil pH was 5.6–6.1, soil organic carbon was 3.24–5.24%, soil total N was 0.17–0.38%, total P was 0.041–0.092%, bulk density was 42–65 g cm$^{-3}$, and C:N ratio varied from 9.8 to 18.2 (Table 1).

**Table 1.** Characteristics of the studied forest stands. Values are site means ($\pm$ SE, when provided).

| Parameter | Stand I (HB) | Stand II (HS) | Stand III (HT) | ANOVA | |
|---|---|---|---|---|---|
| | | Site | | | |
| | | | | F Value | p Value |
| | | **Soil** | | | |
| sand (%) | 70 $^a$ $\pm$ 0.88 | 78 $^b$ $\pm$ 0.88 | 78 $^b$ $\pm$ 0.88 | 48.077 | 0.000 |
| silt (%) | 18 $^b$ $\pm$ 0.88 | 12 $^a$ $\pm$ 0.88 | 12 $^a$ $\pm$ 0.58 | 11.645 | 0.009 |
| clay (%) | 13 $^a$ $\pm$ 1.76 | 11 $^a$ $\pm$ 0.33 | 10 $^a$ $\pm$ 0.33 | 1.837 | 0.239 |
| bulk density (g cm$^{-3}$) | 0.65 $^b$ $\pm$ 0.01 | 0.60 $^b$ $\pm$ 0.09 | 0.42 $^a$ $\pm$ 0.01 | 66.690 | 0.000 |
| WHC (%) | 45.65 $^b$ $\pm$ 0.43 | 42.76 $^{ab}$ $\pm$ 0.61 | 41.47 $^a$ $\pm$ 0.69 | 6.201 | 0.035 |
| moisture (%) | 22.48 $^c$ $\pm$ 0.37 | 14.70 $^b$ $\pm$ 0.19 | 11.36 $^a$ $\pm$ 0.35 | 68.044 | 0.000 |
| porosity | 75.35 $^a$ $\pm$ 0.33 | 77.23 $^a$ $\pm$ 3.28 | 84.28 $^b$ $\pm$ 0.33 | 11.986 | 0.008 |
| pH | 5.67 $^a$ $\pm$ 0.03 | 5.87 $^b$ $\pm$ 0.03 | 6.13 $^c$ $\pm$ 0.03 | 30.659 | 0.001 |
| organic C (%) | 5.24 $^c$ $\pm$ 0.09 | 3.75 $^b$ $\pm$ 0.07 | 3.24 $^a$ $\pm$ 0.04 | 1008.148 | 0.000 |
| total N (%) | 0.38 $^b$ $\pm$ 0.01 | 0.38 $^b$ $\pm$ 0.01 | 0.17 $^a$ $\pm$ 0.01 | 85.809 | 0.000 |
| total P (%) | 0.09 $^c$ $\pm$ 0.00 | 0.06 $^b$ $\pm$ 0.00 | 0.04 $^a$ $\pm$ 0.00 | 324.328 | 0.000 |
| C:N | 13.4 $^a$ $\pm$ 0.17 | 9.8 $^b$ $\pm$ 0.13 | 18.2 $^c$ $\pm$ 0.15 | 743.831 | 0.000 |
| | | **Vegetation** | | | |
| Tree species richness | 05 (*Boehrmeria regulosa, Myrica esculenta, Pinus roxburghii, Quercus leucotrichophora, Rhus valagaris*) | 03 (*Boehrmeria regulosa, Pinus roxburghii, Quercus leucotrichophora*) | 02 (*Pinus roxburghii, Quercus leucotrichophora*) | | |
| Tree density (stems ha$^{-1}$) | 670 | 590 | 570 | | |
| Basal area (m$^2$ ha$^{-1}$) | 32.39 | 27.73 | 27.20 | | |

Different small letters after the mean values in each row represent the significant difference ($p < 0.05$) in trait values following Duncan's post hoc test.

### 3.2. Microbial C, N and P

The microbial C, N and P were recorded maximum was at the HB forest stand and the minimum at the HT forest stand. The microbial biomass C ranged between 730 µg g$^{-1}$ and 751 µg g$^{-1}$ at HB, 718 and 737 µg g$^{-1}$ at HS, and 681 and 697 µg g$^{-1}$ at HT. The values of microbial biomass N were 111 to 143 µg g$^{-1}$ at HB, 93 to 111 µg g$^{-1}$ at HS and 73 to 99 µg g$^{-1}$ at HT. The microbial biomass P was estimated between 53 and 72 µg g$^{-1}$ at HB, 38 and 49 µg g$^{-1}$ at HS, 23 and 36 µg g$^{-1}$ at HT. (Table 2). Similar seasonal variation was recorded at all the stands with maximum values of microbial C, N and P during the rainy season and minimum values during the winter season (Table 2).

**Table 2.** Microbial C, N and P in the soils of forest stand I, II and III ($\mu g\ g^{-1}$ soil $\pm$ S.E.).

|  | Stand I (HB) | Stand II (HS) | Stand III (HT) |
|---|---|---|---|
| Microbial biomass carbon (MBC) |  |  |  |
| Rainy | 751 ± 2.58 | 737 ± 1.02 | 697 ± 0.55 |
| Winter | 730 ± 0.86 | 718 ± 1.32 | 681 ± 1.81 |
| Summer | 738 ± 1.39 | 725 ± 1.18 | 685 ± 2.01 |
| Annual mean | 739.67 ± 6.12 | 726.67 ± 5.55 | 687.67 ± 4.81 |
| Microbial biomass nitrogen (MBN) |  |  |  |
| Rainy | 143 ± 8.54 | 111 ± 4.43 | 99 ± 2.14 |
| Winter | 111 ± 1.78 | 93 ± 1.41 | 73 ± 1.38 |
| Summer | 120 ± 1.45 | 103 ± 1.42 | 89 ± 1.07 |
| Annual mean | 124.67 ± 9.53 | 102.33 ± 5.21 | 87.00 ± 7.57 |
| Microbial biomass phosphorus (MBP) |  |  |  |
| Rainy | 72 ± 3.03 | 49 ± 3.69 | 36 ± 1.86 |
| Winter | 53 ± 1.57 | 38 ± 1.53 | 23 ± 1.55 |
| Summer | 65 ± 1.79 | 45 ± 1.91 | 26 ± 1.73 |
| Annual mean | 63.33 ± 5.55 | 44.00 ± 3.21 | 28.33 ± 3.93 |
| Microbial C:N |  |  |  |
| Rainy | 5.25 ± 0.33 | 6.63 ± 0.26 | 7.04 ± 0.15 |
| Winter | 6.57 ± 0.10 | 7.72 ± 0.13 | 9.33 ± 0.19 |
| Summer | 6.15 ± 0.08 | 7.03 ± 0.07 | 7.69 ± 0.08 |
| Annual mean | 6.16 ± 0.39 | 7.23 ± 0.32 | 8.19 ± 0.22 |
| Microbial C:P |  |  |  |
| Rainy | 10.66 ± 0.40 | 14.31 ± 1.21 | 19.71 ± 0.98 |
| Winter | 13.81 ± 0.41 | 18.41 ± 0.74 | 35.89 ± 2.42 |
| Summer | 11.42 ± 0.33 | 16.43 ± 0.70 | 27.56 ± 1.75 |
| Annual mean | 11.96 ± 0.95 | 16.38 ± 1.18 | 27.72 ± 1.54 |
| Microbial N:P |  |  |  |
| Rainy | 1.98 ± 0.11 | 2.26 ± 0.10 | 2.75 ± 0.18 |
| Winter | 2.09 ± 0.09 | 2.45 ± 0.13 | 3.17 ± 0.28 |
| Summer | 1.85 ± 0.06 | 2.29 ± 0.11 | 3.42 ± 0.19 |
| Annual mean | 1.97 ± 0.04 | 2.33 ± 0.08 | 3.11 ± 0.15 |

The microbial C:N ratio in the present study varied between 5.25 and 9.33, which indicates the dominancy of a fungal community. The microbial biomass C:N ratio increased with increasing altitude (Table 2). The microbial C:P ranged from 10.66 to 35.89. Moreover, the microbial biomass C:P ratio and N:P ratio also showed a similar trend as described for C:N ratio (Table 2).

Two-way analysis of variance (ANOVA) indicated a significant effect ($p < 0.01$) of season and altitude on all the studied parameters; however, the interactive effects of season and altitude showed insignificant variations in the parameters except for C:N, C:P and N:P ratios (Table 3).

**Table 3.** Two-way ANOVA showing effect of altitude, season and their interaction on soil microbial properties in mixed oak–pine forest.

| Variables | Source | Sum of Squares | Df | Mean Square | F | Sig. |
|---|---|---|---|---|---|---|
| MBC | S | 1598.06 | 2 | 799.03 | 113.52 | 0.000 |
| | A | 13274.20 | 2 | 6637.10 | 942.92 | 0.000 |
| | S × A | 25.87 | 4 | 6.47 | 0.92 | 0.474 |
| MBN | S | 2818.66 | 2 | 1409.33 | 38.63 | 0.000 |
| | A | 6575.10 | 2 | 3287.55 | 90.10 | 0.000 |
| | S × A | 237.36 | 4 | 59.34 | 1.63 | 0.211 |
| MBP | S | 1162.40 | 2 | 581.20 | 40.27 | 0.000 |
| | A | 6135.68 | 2 | 3067.84 | 212.58 | 0.000 |
| | S × A | 73.51 | 4 | 18.38 | 1.27 | 0.317 |
| Microbial C:N | S | 10.59 | 2 | 5.30 | 56.13 | 0.000 |
| | A | 18.88 | 2 | 9.44 | 100.06 | 0.000 |
| | S × A | 1.66 | 4 | 0.42 | 4.40 | 0.012 |
| Microbial C:P | S | 330.30 | 2 | 165.15 | 26.51 | 0.000 |
| | A | 1238.75 | 2 | 619.37 | 99.44 | 0.000 |
| | S × A | 218.53 | 4 | 54.63 | 8.77 | 0.000 |
| Microbial N:P | S | 1.28 | 2 | 0.64 | 8.89 | 0.002 |
| | A | 10.18 | 2 | 5.09 | 70.63 | 0.000 |
| | S × A | 1.34 | 4 | 0.33 | 4.65 | 0.009 |

Where, S—Seasons, A—altitude, MBC—microbial biomass carbon, MBN—microbial biomass nitrogen, MBP—microbial biomass phosphorus.

### 3.3. Microbial Biomass and Abiotic Variables

The microbial biomass C, N and P showed significant positive correlations with altitude, silt particles, water-holding capacity, soil moisture, organic carbon, total nitrogen and total phosphorus (Table 4). The microbial biomass C, N and P showed positive correlations with bulk density but were negatively correlated with sand particles, soil porosity and soil pH. The microbial N and P exhibited a significant negative correlation with sand particles and soil pH (Table 4). Soil microbial biomass C and total soil C, soil microbial biomass N and total soil N, and soil microbial biomass P and total soil P were strongly and linearly related (Table 4). However, the present study also indicated that microbial biomass C was strongly associated with soil microbial N (0.80, $p < 0.01$) and P (0.89, $p < 0.01$) content. This strong relationship among all elements indicates that soil microbial biomass C depends on soil N and P to maintain the required microbial element stoichiometry. Soil microbial biomass N and P also showed a strong linear relationship (0.92, $p < 0.01$). The soil-moisture content could be a better indicator of seasonal variation in soil microbial biomass C, N and P as indicated by significant positive correlations between the microbial biomass C, N or P and moisture content (Table 4).

**Table 4.** Correlation coefficients for the relationship of microbial biomass (C, N and P) with altitude and abiotic variables.

| | Altitude | Sand | Silt | Clay | WHC | bD | Mo | Po | pH | C | N | P |
|---|---|---|---|---|---|---|---|---|---|---|---|---|
| MBC | 0.985 ** | −0.720 * | 0.685 * | 0.455 | 0.822 ** | 0.819 ** | 0.916 ** | −0.819 ** | −0.967 ** | 0.897 ** | 0.939 ** | 0.908 ** |
| MBN | 0.963 ** | −0.905 ** | 0.907 ** | 0.496 | 0.928 ** | 0.650 | 0.987 ** | −0.649 | −0.890 ** | 0.992 ** | 0.734 * | 0.992 ** |
| MBP | 0.948 ** | −0.957 ** | 0.898 ** | 0.624 | 0.920 ** | 0.619 | 0.987 ** | −0.619 | −0.884 ** | 0.987 ** | 0.685 * | 0.984 ** |

* Significant at ($p < 0.05$) and ** Significant at ($p < 0.01$).

### 3.4. Multivariate Analyses (PCA) of Microbial Properties of Soil

Principal component analysis (PCA) was carried out to differentiate forest stands by physico-chemical and biological properties of soil (sand, silt, clay, WHC, soil microbial biomass). The multivariate analysis indicated that F1 (active sites with 38.64% variation) and F2 (active variables with 31.24% variation) components exhibited the maximum varia-

tion with physico-chemical and biological properties of soil (Figure 2) and their cumulative variability was about 70% (Table 4).

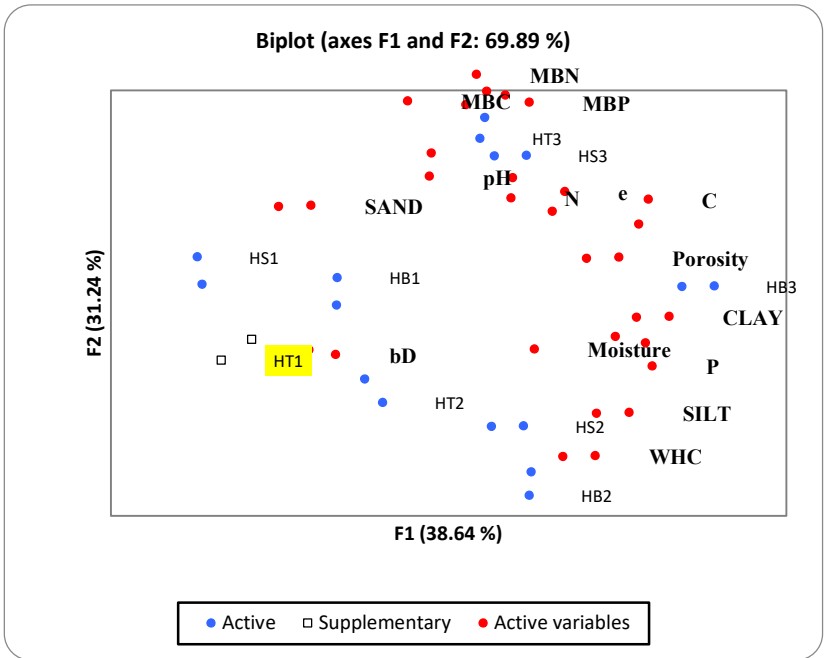

**Figure 2.** Principal component analysis (PCA) of physical, chemical and microbial parameter of soil in different stands of mixed oak–pine forest. PCA axis 1 (38.64%) and 2 (18.66%) represent the first and second coordinates of sites, respectively (bD—bulk density, WHC—water-holding capacity, e—void ratio, C—carbon, N—nitrogen, P—phosphorus, MBC—microbial biomass carbon, MBN—microbial biomass nitrogen, MBP—microbial biomass phosphorus, HT1—Hill top 1, HT2—Hill top 2, HT3—Hill top 3, HS1—Hill slope 1, HS2—Hill slope 2, HS3—Hill slope 3, HB1—Hill base 1, HB 2—Hill base 2, HB3—Hill base 3).

## 4. Discussion

Our values of microbial C were similar to the ranges of 61–2000 mg g$^{-1}$ for various temperate and tropical forest soils [36,37], and 978–2088 mg g$^{-1}$ for sub-tropical forest [38]. Comparatively, microbial N also showed a similar trend with coniferous forest soils reported by Martikainen and Palojarvi [39] as 52–125 mg g$^{-1}$ and evergreen forest soils reported by Diaz-Ravina et al. [40] as 42–242 mg g$^{-1}$, but lower than that of broad-leaved deciduous forest soils [40] (132–240 mg g$^{-1}$). The microbial P values fell well within the reported range of 5.3–67.2 mg g$^{-1}$ for arable land, grassland and woodland soils [34], and 14–46 mg g$^{-1}$ for sub-tropical moist forest reported by Arunachalam and Arunachalam [38]. Several studies have revealed the effects of different systems on microbial communities; however, differences in combinations of variables, such as climate and soil type, have led to different microbial responses in different systems [41,42].

### 4.1. Seasonal Variation in Microbial Biomass

The species composition of the three forest stands were different and this has a significant effect on soil microbial biomass; however, the seasonal pattern of microbial biomass was common at all three sites indicating that the seasonal pattern of soil microbial biomass was regulated by climatic factors. The microbial C, N and P were significantly higher during the rainy season ($p < 0.01$) and lower in the winter season (Table 2). This may be due to higher immobilization of nutrients from the decomposing litter by microbes as decomposition rates of litter and microbial activity are at their peak during the rainy period. Various authors [43–47] have reported that due to high humidity and temperature, growth of microorganisms increased during this season and contributed to the soil micro-

bial biomass. In contrast, for tropical dry deciduous forest, savanna and temperate pastures, Saratchandra et al. [47] and Singh et al. [2] reported maximum values of microbial biomass during summers. Arunachalam and Arunachalam [38] reported maximum values during winters. This may be due to differences in quality of litter and rainfall patterns in these forest types. Low values of microbial C, N and P during the winter season in the present study may be due to low activity of microorganisms and slow rates of decomposition of litter in dry and cool periods. Diaz-Ravina et al. [48] reported that lack of water seemed to limit the microbial biomass more than temperature because lower microbial biomass content was observed in dry periods than in wet periods. The microbial populations altered during the seasons [49]. During the rainy season, increased temperature and moisture significantly promoted the growth of soil microbes [50].

### 4.2. Effect of Altitude

The hypothesis that the decrease in microbial biomass with increase in altitude in all the seasons in central Himalayan mixed oak–pine forest was partly demonstrated in the present study. During all the sampling seasons, microbial C, N and P decreased with the increase in altitude (Table 2). Koch et al. [51] and Liu and Wang [52] also reported that microbial biomass decreased with increasing altitude. According to Lipson et al. [1] at low altitude, snow cover provides a protective effect during winter and before the beginning of the growing season snow melt and leaching of nutrients from high altitudes provide rich substrate inputs to low altitudes and results in higher microbial activity. Higher soil-moisture content (Table 1), less exposure to sunlight, higher plant diversity [45] and better quality of litter at HB resulted in better growth of microbes. Patel et al. [53] also reported that microbial C, N and P were comparatively higher in forest stands situated at the foothill than in forest stands located at higher elevation. However, Wardle [54] reported that there are no consistent seasonally determined temporal patterns of microbial biomass change in tropical and warm temperate ecosystems. With increasing altitude, the ratios of microbial C, N and P to total soil organic C, total N and P increased. This indicates that microbial biomass/nutrients (C, N and P) more frequently immobilized at high altitude, i.e., the HT stand.

### 4.3. Microbial Quotient

The microbial C:N ratios (5.25–9.33) reported in the present study are similar to the range reported by Martikainen and Palojarvi [39] for various forest soils (6–9). The C:N ratio of fungi is often 10–12 and that of bacteria is usually between 3 and 5. Because C:N ratios in the present study are more than 5, the soils may be dominated by fungal communities. In the present study, microbial biomass C:N ratios increased with increasing altitude (Table 2). Arunachalam and Pandey [55] stated that microbial C:N ratio is an indicator of ecosystem recovery as the lower the ratio, the shorter will be the time required for build-up of the microbial population and their activity. An increase in microbial biomass C:N ratio is considered an indication of changes in the microbial community, with the possible dominance of fungi over bacteria; thus, it could be suggested that the soil at HT is fungi-dominated as compared to HB and HS. It is possible that restoration of the soil at HT would take much longer than restoration of the soil at HB as apparent from the lower C:N ratio in the HB than in HS and HT (Table 2). The microbial C:P ratio in the present study (10.66–35.89) falls well within the reported range of 10.6–35.9 by Brookes et al. [34], but lower than the sub-tropical humid forest (33.2–98.5) reported by Arunachalam and Arunachalam [38], which may be due to high microbial biomass P in the present forest. Moreover, the microbial biomass C:P ratio and N:P ratio also showed similar trends as described for C:N ratio (Table 2). Soil microbial quotient increased with increasing altitude representing the occurrence of a less active nutrient pool in soils with increasing altitude [56].

*4.4. Relationship between Microbial Biomass and Abiotic Variables*

Comparing different soil types, topography, and drainage is primarily significant when assessing their impacts on soil microbial communities because microbial populations are drastically influenced by physical and chemical soil properties [57]. In the central Himalayan region, the availability of soil moisture depends on rainfall. Any change in the rainfall pattern may have an impact on the soil microbial biomass dynamics, which in turn would influence C, N and P cycling in the region. Similar reports related to significant positive correlation between soil moisture and soil microbial biomass in wet tropical deciduous forests of India have also been reported by Devi and Yadava [44]. Microbial biomass of soil is significantly influenced by soil organic matter [56–59]. Results of the present study corroborate previous research showing strong correlations among microbial biomass, soil C and nutrient availability [54]. Microbial composition and their abilities respond to short-term changes in soil such as inorganic N availability as well as long-term changes like SOM [60].

## 5. Conclusions

The high value of microbial biomass carbon, nitrogen and phosphorus in the hill base forest stand as compared to other high-altitude forest stands indicated that this region had a higher number of microorganisms which sustain better soil quality due to the more diverse forest structure, higher soil-moisture content, better litter quality and less exposure to sunlight. This site was also enriched with substrate input through rainwater, snowmelt and leaching of nutrients from the hill top which favoured high microbial activity. At each altitude, soil microbial biomass showed similar seasonal variation with a minimum value during the winter season and maximum value during the rainy season, which can be considered a nutrient conservation strategy. The soil microbial biomass C, N and P exhibited significant negative correlations with altitude ($p < 0.05$). Several soil parameters such as silt, water-holding capacity, moisture content, and total C, N and P showed positive correlations, while sand, porosity and pH showed significant negative correlations with soil microbial biomass C, N and P. Microbial C:N ratio in the present study revealed the dominance of fungal communities over bacterial communities across all altitudes. The microbial C:N, C:P and N:P ratios increased with increasing altitude indicating that soil fertility is influenced by the species composition of the forest stand.

**Author Contributions:** V.M. performed the experiments and analysed the data. K.B. wrote and revised the manuscript. S.S.B. guided the research and modified the manuscript. H.K. and R.K.C. provided editorial advice and finally revised the paper. All authors have read and agreed to the published version of the manuscript.

**Funding:** The necessary lab facilities provided by the Head, Department of Botany, Kumaun University and Tea Development Board, Bhowali, Nainital are highly acknowledged. R.K.C. thanks the National Natural Science Foundation of China (NSFC), Chinese Academy of Science, China (award No. 31750110466) for financial support.

**Institutional Review Board Statement:** Not applicable.

**Informed Consent Statement:** Informed consent was obtained from all subjects involved in the study.

**Data Availability Statement:** The data used during the current study are available from the corresponding author on reasonable request.

**Acknowledgments:** The authors thank the three anonymous reviewers for their constructive comments for improving the standard of the manuscript.

**Conflicts of Interest:** The authors declare no conflict of interest.

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
