# Peer review of "Seasonal Dynamics of Soil Microbial Biomass C, N and P along an Altitudinal Gradient in Central Himalaya, India"

_sustainability, doi:10.3390/su15021651_

Round 1
Reviewer 1 Report
Dear Authors,
you have presented interesting research on soil biology and the seasonal variation of C, N and P concentrations in the soil. In my opinion, two-year research is too little to be able to form constructive and factual conclusions in agriculture, Experiments are conducted for 3 years, because then the average of the conditions of the research season can be more similar to typical conditions.
The paper is well written and has an interesting discussion of the results and conclusions.
I suggest that the work be resubmitted: a) adding another year of research, or b) as a communication/pilot study.
Author Response
Comment 1: You have presented interesting research on soil biology and the seasonal variation of C, N and P concentrations in the soil. In my opinion, two-year research is too little to be able to form constructive and factual conclusions in agriculture, Experiments are conducted for 3 years, because then the average of the conditions of the research season can be more similar to typical conditions.
The paper is well written and has an interesting discussion of the results and conclusions. Authors’ response: We are thankful for your appreciation and kind consideration.
We agreed with your suggestions regarding the time duration of 3 years for constructive and factual conclusions in agriculture systems but the present research was conducted in the natural forest of temperate region of Kumaun Himalaya not in the agriculture systems (where the management play a main role in the distribution of microbes), and the main focus of the study was to assess the seasonal and altitudinal changes on the soil microbes in the natural forest ecosystem.
Specific comments
Comment 2: I suggest that the work be resubmitted: a) adding another year of research, or b) as a communication/pilot study.
Authors’ response: The present study could be considered as a pilot research on this topic in the region of Uttarakhand.

Reviewer 2 Report
The manuscript entitled "Seasonal Dynamics of Soil Microbial Biomass C, N and P Along an Altitudinal Gradient in Central Himalaya, India concerns the study of the seasonal dynamics at an altitude of understanding the soil microbial activity in oak pine forests in Central Himalaya. The subject of the manuscript falls within the general scope of the Journal and provides interesting data. However, the paper shows several gaps and weaknesses that need to be addressed. Below are some comments:
Abstract
In my opinion, it is written well, but needs the incorporating some data on the effects of the season on microbial activity in oak pine forests.
Introduction
Incorporate the suggestion given in the manuscript for improving the knowledge of the scientific community.
Materials and Methods
Incorporate the highlighted comments in the materials and methods section for the soundness of the study.
Results and discussion
This section of the manuscript is written well, but the references incorporated in the manuscript are old, incorporate some new. In my view, results of the effects of season on microbial activity.

Author Response
Comment 1: The manuscript entitled "Seasonal Dynamics of Soil Microbial Biomass C, N and P Along an Altitudinal Gradient in Central Himalaya, India concerns the study of the seasonal dynamics at an altitude of understanding the soil microbial activity in oak pine forests in Central Himalaya. The subject of the manuscript falls within the general scope of the Journal and provides interesting data. However, the paper shows several gaps and weaknesses that need to be addressed.
Authors’ response: We are grateful for your appraisal and suggestions.
The suggested modifications have been done to improve the quality of the article.
Specific Comments
Comment 2: Abstract: In my opinion, it is written well, but needs the incorporating some data on the effects of the season on microbial activity in oak pine forests.
Authors’ response: The data related with the seasonal effects on the microbial activity have been added to the abstract section as suggested.
Comment 3: Introduction: Incorporate the suggestion given in the manuscript for improving the knowledge of the scientific community.
Authors’ response: The suggestions have been incorporated in the Introduction section accordingly.
Comment 4: Line No. 36 Add reference.
Authors’ response: The reference has been added now.
Comment 5: Line No. 60-65 rewrite
Authors’ response: The sentence of line no. 60-65 is reframed, as suggested.
Comment 6: Materials and Methods: Incorporate the highlighted comments in the materials and methods section for the soundness of the study.
Authors’ response: The highlighted comments have been incorporated in the Materials and Methods section.
Comment 7: Line No. 68 delete.
Authors’ response: Deleted the repetition of “in the” in line no.68.
Comment 8: Line No. 72 full name?
Authors’ response: The full names of abbreviations used for HB, HS and HT mentioned in line no. 72.
Comment 9: Line No. 73 Only two hundred m attitudinal difference the variation in shrub species can be possible?
Authors’ response: In the Himalayan region, due to the change in topology and soil microclimatic conditions at the short distances, it is possible that the under canopy vegetation changed and dominates within short distances and altitudes.
We have also reframed the sentence of line no. 76-82 now.
Comment 10: Results and discussion: This section of the manuscript is written well, but the references incorporated in the manuscript are old, incorporate some new. In my view, results of the effects of season on microbial activity.
Authors’ response: Thank you for the appreciation. The changes have been made and the new references also included in the Discussion section as per the suggestions.
Comment 11: Table No. 1 name of tree species.
Authors’ response: The names of the tree species have been added in table no. 1.
Comment 12: Table No. 2 value of SD?
Authors’ response: We have incorporated the SD values in table no. 2.
Comment 13: Two way analysis of variance may be appropriate?
Authors’ response: We have performed Two way ANOVA considering system and altitude as two independent variables and the output has been included in the result section.

Reviewer 3 Report
Title : Seasonal Dynamics of Soil Microbial Biomass C, N and P Along an Altitudinal Gradient in Central Himalaya, India
1. Content
The authors explored changes in microbial variables in forest, as the effects of altitude and season. They found that microbial biomass C, N and P decreased significantly with increasing altitude. Also, they found that maximum values were observed during rainy season, and minimum values during winter. Reversely, microbial C/N ratio was lower in rainy season, which the authors considered as a strategy for soil microorganisms to conserve nutrient.
2. Reviewer main comments
The subject is important, particularly knowing that carbon and nutrient cycling are primarily processed by soil microorganisms. Dealing with the spatial or temporal dynamics of MBC in natural ecosystems such as forest could potentially inform management strategy to reduce CO2 release into atmosphere.
Unfortunately, this MS, in its current form holds many defaults so that it could not be recommended for publication.
In fact, the authors failled in specifying the main objective of the work ; and as a consequence, the relevance of the work could not be appreciated. Definitely, data must be subjected to a two-way Anova which will indicate whether there is a « forest type x altitude » interaction effect on the studied microbial variables. The results are not presented concomitently with the statistical results, the significance of the effets tested could not be appreciated. Strangely, some results are presented in the discussion section. All these indicate that the manuscript was not sufficiently ready for submission.
3. Specific comments
Abstract
Main objective not stated. The relevance of the correlations among the microbial variables was not explained. The authors wrote « microbial C, N and P showed positive correlation with soil bulk density and porosity… » This should be checked since bulk density and porosity are negatively correlated. If the microbial variable are positively correlated with BD, then they should be negatively correlated with porosity…
Material and Methods
L70 : Please clarify, natural forest or planted forest ?
Results
For all the section, comparisons among sites should be conducted and p-values shown, either in the text or in Tables please. More importantly, the authors should perform two-way ANOVA on the data as the soil variables were subjected to the effects of two factors, sites and seasons. Show if there is a significant interaction of the factors or not.
Table 2 : The three later parameter were not mentioned as studied parameters in the M&M section. Their importance in the study wasnote explained.
Heading 3.3 : Unfortunately, this way is not the rigth one to show the effets of forest altitude or seasons! Comparison should be made at the altitude level and the season level (in case there no significant interaction between the two factors).
L155-157 : The results of statistical tests should accompany each soil parameter considered, instead of encompassing them into one paragraph at the end of the Results section. This way is not acceptable.
Table 3 : What about the other microbial variables and ratios?
Altitude was not a parameter influenced by forest site or seasons. I cannot understand the presence of this parameters in Table 3 !
Discussion
The heading 4.4 Heavily reports the results of the correlations tests, this is not correct as we are in the Discussion section. The authors must rather explain these correlations.
Heading 4.4 and 4.5 Must be reformulated as they reflect results in their current form. These forms are rather suited for the Results section, not the Discussion section.
Heading 4.5. Multivariate analyses (PCA) of microbial properties of soil : Why some results are presented in the Discussion section ? PCA and Correlations ?
Conclusion
Very weak ! This is not surprising as the main objective of the study was not stated, nor the hypotheses !
Author Response
Comment 1: The subject is important, particularly knowing that carbon and nutrient cycling are primarily processed by soil microorganisms. Dealing with the spatial or temporal dynamics of MBC in natural ecosystems such as forest could potentially inform management strategy to reduce CO2 release into atmosphere.
Unfortunately, this MS, in its current form holds many defaults so that it could not be recommended for publication.
In fact, the authors failed in specifying the main objective of the work ; and as a consequence, the relevance of the work could not be appreciated. Definitely, data must be subjected to a two-way Anova which will indicate whether there is a « forest type x altitude » interaction effect on the studied microbial variables. The results are not presented concomitantly with the statistical results, the significance of the effects tested could not be appreciated. Strangely, some results are presented in the discussion section. All these indicate that the manuscript was not sufficiently ready for submission.
Authors’ response: Authors are highly obliged for your valuable suggestions to improve the manuscript.
Hope you would be agreed with the revised version of the manuscript, as this improved manuscript included all the suggestions of all the reviewers.
Specific Comments
Comment 2: Abstract: Main objective not stated. The relevance of the correlations among the microbial variables was not explained. The authors wrote « microbial C, N and P showed positive correlation with soil bulk density and porosity… » This should be checked since bulk density and porosity are negatively correlated. If the microbial variables are positively correlated with BD, then they should be negatively correlated with porosity…
Authors’ response: The main objectives are now stated in the abstract also. The relevance of the correlations among the microbial variables is explained now. The sentence related with the relation between microbial variables and bulk density has been corrected.
Comment 3: Material and Methods: L70 : Please clarify, natural forest or planted forest ?
Authors’ response: It is a natural forest. Now cleared both in the abstract as well as Material and Method sections.
Comment 4: Results: For all the section, comparisons among sites should be conducted and p-values shown, either in the text or in Tables please. More importantly, the authors should perform two-way ANOVA on the data as the soil variables were subjected to the effects of two factors, sites and seasons. Show if there is a significant interaction of the factors or not.
Authors’ response: As per your suggestion, the statistical analysis was performed in between the site data and the results have been included in the tables. In addition, to check the interaction of seasons and altitudes, two-way ANOVA results has also been added in the results section.
Comment 5: Table 2: The three later parameter were not mentioned as studied parameters in the M&M section. Their importance in the study was not explained.
Authors’ response: The said parameters have been deleted from the entire manuscript as they were not detailed in the article.
Comment 6: Heading 3.3: Unfortunately, this way is not the right one to show the effects of forest altitude or seasons! Comparison should be made at the altitude level and the season level (in case there no significant interaction between the two factors).
Authors’ response: Two-way ANOVA results are included in the manuscript (Results section) to show the effects of forest altitudes, seasons and their interactive effects.
Comment 7: L155-157: The results of statistical tests should accompany each soil parameter considered, instead of encompassing them into one paragraph at the end of the Results section. This way is not acceptable.
Authors’ response: The results have been modified now with the proper incorporation of statistical tests.
Comment 8: Table 3: What about the other microbial variables and ratios? Altitude was not a parameter influenced by forest site or seasons. I cannot understand the presence of this parameters in Table 3 !
Authors’ response: Table 3 has been improved including all the microbial variables and ratios as per your suggestion.
Comment 9: Discussion:The heading 4.4 Heavily reports the results of the correlations tests, this is not correct as we are in the Discussion section. The authors must rather explain these correlations.
Authors’ response: Heading 4.4 shifted from discussion section to the result section now as under the heading 3.3.
Comment 10: Heading 4.4 and 4.5 Must be reformulated as they reflect results in their current form. These forms are rather suited for the Results section, not the Discussion section.
Authors’ response: Heading 4.4 and 4.5 shifted from discussion section to the result section now as under the heading 3.3 and 3.4.
Comment 11: Heading 4.5. Multivariate analyses (PCA) of microbial properties of soil : Why some results are presented in the Discussion section ? PCA and Correlations ?
Authors’ response: The Multivariate analyses (PCA) of microbial properties of soil is now presented in results under the heading 3.5 as per your suggestion.
Comment 12: Conclusion- Very weak! This is not surprising as the main objective of the study was not stated, nor the hypotheses!
Authors’ response: The conclusion is reframed now and concluded the objective and hypothesis as per your suggestions.
We are highly thankful to you for the valuable suggestions and closer look which have improved our manuscript.

Round 2
Reviewer 1 Report
Your arguments convinced me, the research was conducted under specific conditions and terrain. I am willing to accept the paper because the authors have made a great contribution to improving the manuscript.